# Delay-Aware Reinforcement Learning: Insights From Delay Distributional Perspective

## Abstract

Although deep reinforcement learning (DRL) has achieved great success across various domains, the presence of random delays in real-world scenarios (e.g., remote control) poses a significant challenge to its practicality. Existing delay-aware DRLs mainly focus on state augmentation with historical memory, ensuring that the actions taken are aligned with the true state. However, these approaches still rely on the conventional expected $Q$ value. In contrast, to model delay uncertainty, we aim to go beyond the expected value and propose a distributional DRL to represent the distribution of this $Q$ value. Based on the delay distribution, we further propose a correction mechanism for the distributional $Q$ value, enabling the agent to learn accurate returns in delayed environments. Finally, we apply these techniques to design the delay-aware distributional actor-critic (DADAC) DRL framework, in which the critic is the corrected distributional value function. Experimental results demonstrate that compared to the state-of-the-art delay-aware DRL methods, the proposed DADAC exhibits substantial performance advantages in handling random delays in the MuJoCo continuous control tasks. The corresponding source code is available at https://anonymous.4open.science/r/DADAC.

## 1 Introduction

Deep Reinforcement Learning (DRL) has demonstrated substantial success across diverse domains and has become a prominent focus in AI research. Its capacity to learn optimal policies in dynamic environments has catalyzed breakthroughs in areas such as gaming, robotics, autonomous systems, and finance. Noteworthy accomplishments include surpassing human expertise in games like Go and Atari, enhancing robotic control, and improving decision-making processes across industries.

However, applying reinforcement learning to real-world scenarios presents numerous challenges, with delay being one of the most significant issues (Dulac-Arnold et al., 2021). Conventional reinforcement learning follows Markov Decision Process (MDP) and assumes that the interaction between the agent and the environment is instantaneous; that is, the action taken by the agent immediately affects the environment, and the agent receives prompt feedback from the environment (e.g., observations, rewards, etc.). In practical systems, however, this ideal is often compromised by real-world delays, due to the limited communication bandwidths or constrained computational resources. For instance, self-driving cars experience computational and perceptual delays when sensing the environment through various cameras and sensors (Strobel et al., 2020). Teleoperation tasks are affected by network delays, while robot control tasks encounter execution delays in power and mechanical systems (Kebria et al., 2019). In real-world scenarios, delay always exists and thus the assumption of immediate interaction between the agent and environment does not hold, as the feedback that the agent receives at any given time may correspond to a prior timestep due to delays. This discrepancy renders conventional DRL less efficacious in environments characterized by delays. The challenge is even more severe when these delays are random and unpredictable.

To address this issue, the most common approach is state augmentation, which incorporates historical action information into the original state to ensure that the MDP framework remains applicable in delayed environments (Chen et al., 2021; Nath et al., 2021). However, the augmented state space grows exponentially with the number of delayed timesteps, making this approach unsuitable for applications involving large delays. To alleviate the curse of dimensionality associated with state augmentation methods, state prediction methods have been proposed. These methods typically take

historical observations and action information as inputs, and then output predicted observations of the real environment or representations of observations in the form of embeddings (Yu et al., 2023; Liotet et al., 2021). However, in complex dynamic environments, the prediction error of these methods can increase significantly, further complicating the training process. More importantly, most of the aforementioned existing studies (Kim et al., 2023; Yu et al., 2023; Chen et al., 2021) mainly focused on environments with *fixed delays*, which are known in advance, but overlooked the fact that, in real-world applications, delays are often random and unpredictable.

More recently, a few approaches were proposed to handle the *random delays*. Bouteiller et al. (2021) proposed the Delay-Correcting Actor-Critic algorithm by designing the partial trajectory resampling method, which converts off-policy sampled trajectories into on-policy ones in random delay environments. However, this method relies on the observation and action delay values at each timestep and suffers from training inefficiency due to the recursive nature of the partial trajectory resampling process. Wang et al. (2024) recovered delay-free trajectories by time-calibrating historical data and designed a series of state augmentation and prediction-based methods to address the signal delay problem. However, these methods suffer from significant performance degradation in non-fixed delay environments, despite knowing the maximum of delay, due to limitations in the accuracy of the oracle state representation.

In this paper, we aim to address the challenges of deep reinforcement learning applied to the environment with random delays. First, we propose a new value function correction process tailored for random delay environments from the perspective of value correction. This method determines the influence range and probability of an observation or action through delay distribution, thereby accurately reconstructing the value representation of a state-action pair in random delay environments, improving both performance and adaptability of reinforcement learning with random delays. Next, we use distributions rather than expectations to represent value function, allowing for a more precise expression of the value function and a more stable training process in complex, dynamical environments with random delays. Finally, we demonstrate that the proposed method can simultaneously handle both observation delays and action delays.

It is worth noting that our method does not modify the state space, and the agent only receives original observations from the delayed environment. This not only enhances the method's adaptability across different scenarios but also minimizes the introduction of inaccurate information and reduces the extra computational overhead associated with state space modifications. To the best of our knowledge, our proposed method is the first to use delayed original observations as state inputs without modification and to introduce the concept of distributional reinforcement learning in research on addressing reinforcement learning with random delays. This combination allows for more accurate modeling of uncertainty in delayed environments, offering a novel perspective on effectively addressing random delays.

## 2 RELATED WORK

**Delay-aware Reinforcement Learning.** Due to the presence of delays in the environment, the Markov decision process (MDP) becomes inapplicable, leading to the development of variants such as delay-aware MDP and constant delayed MDP for delayed environments. To treat these variants as standard MDP, the state augmentation method is proposed and widely used for fixed delay environments, constructed by concatenating the last observed state with the actions taken since the last visit to that observed state (Chen et al., 2021; Derman et al., 2021). While some research has proposed state augmentation-based approaches for use in random-delay environments, this potential representation of real-time states still faces accuracy limitations (Wang et al., 2024; Nath et al., 2021). Furthermore, the size of the augmented state typically correlates with the delayed steps, causing the size of augmented state space to grow exponentially with the number of delayed steps.

To address this issue, several state prediction-based methods have been proposed to predict the true state in the current real-time environment by utilizing historical state and action information (Wang et al., 2024; Yu et al., 2023; Liotet et al., 2021). However, in complex stochastic delay environments, the accuracy and generality of these prediction methods significantly limit their broader application. Additionally, Kim et al. (2023) proposed a novel belief projection method that tackles the state-space explosion problem by projecting the augmented state space into a smaller one. However, this method is only applicable to fixed-delay environments, which do not reflect real-world practical features.

**Distributional Reinforcement Learning.** Conventional reinforcement learning generally optimises the expectation of the return, but the presence of randomness between the agent and the environment results in the return obeying a distribution given a policy $\pi$. Building on this insight, Bellemare et al. (2017) first introduced the distributional DQN, which represents a return as a discrete distribution of length 51, known as C51. Subsequently, several approaches have been proposed to describe distributions more accurately, providing a solid theoretical and practical foundation for the field of distributional reinforcement learning (Dabney et al., 2018b;a; Rowland et al., 2019; Zhou et al., 2020). Recently, the distributional perspective has also been applied to the actor-critic framework. Nam et al. (2021) proposed the Gaussian Mixture Actor-Critic (GMAC), which models the return as a mixture Gaussian distributions. Specific algorithms, such as the Distributed Distributional Deep Deterministic Policy Gradient algorithm(D4PG) (Barth-Maron et al., 2018) and Distributed Soft Actor-Critic (DSAC) (Duan et al., 2021), have been introduced to address value estimation errors, enhancing the algorithms' value estimation capabilities in complex scenarios. In this work, we introduce the concept of distributional reinforcement learning to enhance the modeling of uncertainty in random delay environments. In contrast to the existing methods, we make further steps in addressing the issue of inaccurate return estimation in environments with random delays.

## 3 PRELIMINARIES

In practical environment, observations and actions can face significant delay due to the constrained communication channel or un-negligible computation time. When the environment provides the feedback of state, the agent may not be able to observe it immediately. The observation may delay for multiple timesteps. Similarly, when the agent makes an action, it may not immediately interact with the environment. The action might be implemented after multiple timesteps, as illustrated in Figure 1. For the environment with random delays, this issue becomes even more critical, as the agent may receive observations from multiple or zero previous timesteps at the current time. This leads to rounding or duplication of observations and actions in the process.

Conventional reinforcement learning is generally modeled as a Markov decision process represented by a 5-tuple $(S, A, R, P, \gamma)$, where $S$ is the state space, $A$ is the action space, $R : S \times A \mapsto \mathbb{R}$ is the reward function, $P : S \times A \times S$ is the transition probability, and $\gamma \in (0, 1)$ is the discount factor. The Bellman equation is utilized to describe the value function as shown below:

$$Q(s, a) = r(s, a) + \gamma \cdot Q(S', A'), \quad (1)$$

where $S' \sim P(\cdot|s, a)$ and $A' \sim \pi(\cdot|S')$. However, in the environment with random delays, the agent receives reward $r_t$ from the environment that does not correspond to the current state-action pair $(s_t, a_t)$, but rather relates to some earlier state-action pair. Therefore, the return corresponding to $(s_t, a_t)$ should be calculated from the time when the agent receives the feedback from the environment. Thus, reusing the Bellman equation in delayed environment for calculating value function would significantly mislead the update of the value function and affect reinforcement learning performance.

Figure 1: An example of remotely controlled UAV navigation. The red mark represents the UAV's state and action in the real-world, i.e. environment with delays, while the blue mark indicates the corresponding state and action in the ideal environment without delays.

To address this challenge, most existing methods implicitly describe the true state in the environment through *state augmentation techniques*, which usually assume that the maximum value of the delay (Wang et al., 2024; Nath et al., 2021) is known in random delay environments, or the true value of the delay is known in either the random delay environments (Bouteiller et al., 2021), or the fixed delay environments (Chen et al., 2021). However, such approaches face two main issues. Firstly, they experience a computational burden that scales exponentially with increasing delay. Secondly, due to the limited representation of the true state, they fail to adequately capture the nuances of random delays, which negatively impacts the training efficacy. *So how to accurately and adaptively implement value function learning in environment with random delays remains an open problem.*

In this study, we gain novel insights into the nature of random delays as shown below. By stepping outside the local horizon of policy updates based on sampled sequential trajectories, we observe that, due to random delays, when the environment feeds back an observation, the agent may receive it with a certain probability at various future timesteps. Similarly, when an agent takes an action, it may interact with the environment at various future timesteps with a certain probability. Considering the probabilistic distribution of delays, it becomes evident that both observations and actions have the potential to influence a spectrum of future time steps with varying degrees of probability. *Consequently, the accurate computation of returns should encompass all the timesteps that might be impacted by these actions and observations, rather than being confined to the immediate time step.* This broader temporal consideration is crucial for a more precise and robust learning process in environments characterized by random delays.

Based on the above insights, in this paper, we assume we are given the probability distributions of delays[1], and derive the delay-aware distributional actor-critic (DADAC), a Soft Actor-Critic-based algorithm to maintain excellent performance in environments with random delays.

## 4  DELAY-AWARE DISTRIBUTIONAL VALUE FUNCTION AND VALUE CORRECTION

In this section, we first introduce the distributional value function that will lead to more stable learning in delayed environments. We then present a value correction mechanism for this distributional value, which can correctly represent the state-action returns in all timesteps that might be impacted due to the random delays.

**Distributional Value Function.** In contrast to the common approach to RL which models the expectations of the return, the distributional value function models the distribution of returns (Bellemare et al., 2017). Distributional perspective preserves multimodality in value distributions, which improves learning stability and thus has been subordinated to specific purposes such as implementing risk-aware behaviour Morimura et al. (2010). Being aware of the representation capability of value distributions, this paper is the first to introduce the distributional value function in delayed environments. Formally, in DRL, let $Q(s, a)$ denote the expectation of the state-action value, the distributional value function can be expressed as $Z(s, a)$ whose expectation is the value $Q(s, a)$, i.e. $Q(s, a) = \mathbb{E}[Z(s, a)]$. This distributional value can be also described by a recursive Bellman equation, but in a distributional manner.

$$Z(s, a) = \sum_{s' \sim P(\cdot|s,a), a' \sim \pi(\cdot|s')} \left[ r(s, a, s') + \gamma \cdot Z(s', a') \right]. \tag{2}$$

In this work, we choose the Gaussian distribution that is highly expressive to approximate the distributional Bellman optimality operator $Z(s, a)$. The goal of our optimization is to achieve a more accurate representation of the Gaussian distribution. The neural network designed to estimate this distribution outputs both the mean and standard deviation, with the Kullback-Leibler (KL) divergence serving as the loss function to measure the difference between the estimated distribution and the true distribution.

**Delay-Aware Value Correction.** In delayed environments, there exists misalignment between the observed and true state-actions. Let us take the action delay as an example. At time step $t$, let $(s_t, a_t, r_t, s_{t+1})$ denote the observed state $s_t$, the action taken $a_t$, the subsequent environmental feedback $s_{t+1}$ as well as the corresponding reward $r_t$. The presence of the action delay implies that the current state-action $(s_t, a_t)$ might not determine the subsequent state $s_{t+1}$. Instead, the agent receives the feedback of $(s_t, a_t)$ after $\Delta t$ delayed timesteps, where $\Delta t$ follows the action delay distribution $D^a$. The reward $r_t$ does not correspond to the state-action $(s_t, a_t)$, and the true reward of $(s_t, a_t)$ is delayed to be obtained at $t + \Delta t$, denoted as $r_{t+\Delta t}$. Therefore, due to the action delay, the return for $(s_t, a_t)$ should be calculated starting from $t + \Delta t$. To this end, we propose value correction mechanisms that aim to recover accurate value functions in random delay environments.

As shown in Figure 2, from the delay distribution perspective, an action may affect multiple timesteps in the future with certain probabilities. Therefore, the estimation of the value function

---

[1]The observation and action delays, which are mainly caused by limited communication bandwidths, typically follow some stochastic patterns (Krasniqi et al., 2020; Wang et al., 2011; Xia & Tse, 2006).

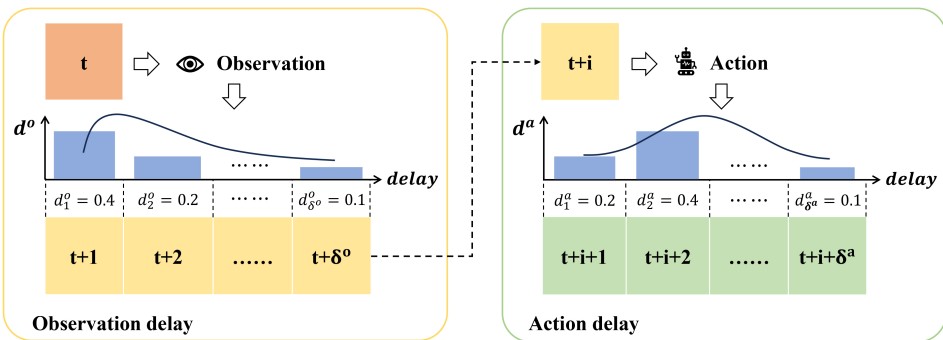

Figure 2: Examples of the distributions of random delays on observations and actions.

$Z(s_t, a_t)$ should account for the probabilistic effect of this delay distribution. Specifically, $Z(s_t, a_t)$ should be associated with the returns between times $t + 1$ and $t + \delta$, where we consider that when the probability of a delay falls below a certain threshold, its impact is negligible. We define the effective delay range as all delays in the distribution that exceed this threshold, with $\delta$ representing the maximum of this range.

$$Z(s_t, a_t) = \sum_{i=1}^{\delta} p_i \cdot \left[ r_{t+i}(s_t, a_t, s_{t+i}) + \gamma^i \cdot \sum_{a_{t+i} \sim \pi(\cdot|s_{t+i})} Z(s_{t+i}, a_{t+i}) \right]. \tag{3}$$

In Eq. (3), $p_i$ denotes the probability that the action delay is $i$ timesteps, $(s_{t+i}, a_{t+i})$ denotes the state received and action taken by the agent at the $i$th step, and $\pi$ is the policy network. The following theorem proves the convergence of Eq. (3).

**Theorem 1** (Convergence of Distributional Value Correction Bellman Equation). *The value correction iteration process in Eq. (3), which map the state-action pair $(s, a)$ to a distributional return in delayed environments, will converge to a unique fixed value as $t \to \infty$. (Proof in Appendix).*

**Unifying Observation and Action Delays.** Real-world scenarios may involve diverse delays, which can significantly increase the problem's complexity. For the case where both observation delay and action delay are considered, we decompose the process based on Eq. (3). As shown in Figure 2, affected by the observation delay, the true state at the time step $t$, will be observed at $t + i$ with the probability of $p_i^o$. At time step $t + i$, given the observation $s_{t+i}$, the agent takes an action $a_{t+i}$ and receives the the environmental feedback (e.g., reward) at the $(t + i + j)$th time step with probability of $p_j^a$. Based on the entire delay probability distribution, we derive the following Eq. (4) to correct the value function considering both observation delay and action delay.

$$\begin{aligned}
Z(s_t, a_t) &= p_1^o \cdot Z(s_{t+1}, a_{t+1}) + \cdots + p_{\delta_o}^o \cdot Z(s_{t+\delta_o}, a_{t+\delta_o}) \\
&= \sum_{i=1}^{\delta_o} p_i^o \cdot (p_1^a \cdot Z(s_{t+i+1}, a_{t+i+1}) + \cdots + p_{\delta_a}^a \cdot Z(s_{t+i+\delta_a}, a_{t+i+\delta_a})) \\
&= \sum_{i=1}^{\delta_o} p_i^o \cdot \sum_{j=1}^{\delta_a} p_j^a \cdot Z(s_{t+i+j}, a_{t+i+j}) \\
&= \sum_{i=1}^{\delta_o} p_i^o \cdot \sum_{j=1}^{\delta_a} p_j^a \cdot (r_{t+i+j} + \gamma^{i+j} \cdot \sum_{a_{t+i+j} \sim \pi(\cdot|s_{t+i+j})} Z(s_{t+i+j}, a_{t+i+j})),
\end{aligned} \tag{4}$$

where $\delta_o$ and $\delta_a$ denote the maximum values of effective observation delay range and effective action delay range, respectively, while $p_i^o$ and $p_i^a$ represent the probabilities that the observation delay and action delay are equal to $i$, respectively. The following theorem shows that it is legal to deal with different delays in the same method.

**Theorem 2** (Equivalence of different delays). *Different variants of delays, including observation delays and action delays, exert equivalent effects on the value correction method. In other words,*

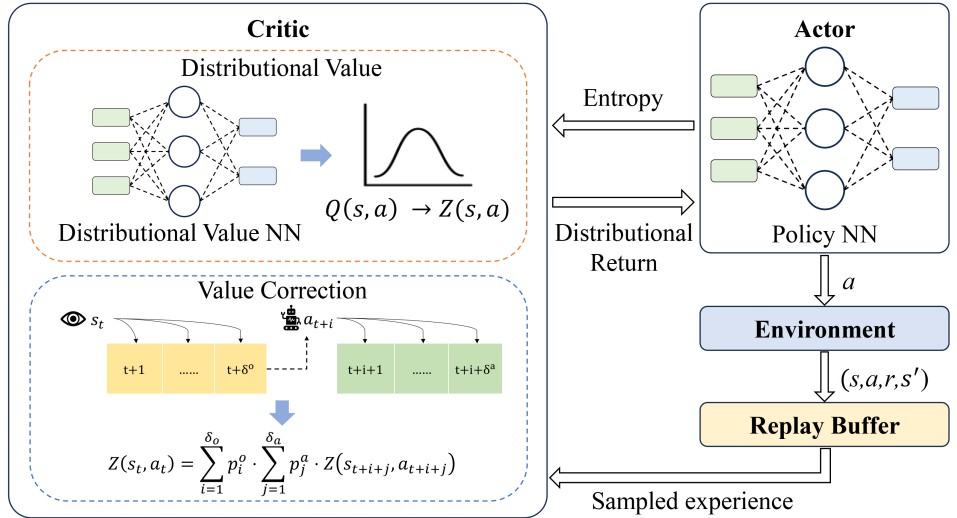

Figure 3: Framework diagram of DADAC.

*the agent is capable of perceiving variants of delays in the same manner, as long as the delay distribution can be accurately characterized. (Proof in Appendix).*

Although the proposed method is primarily designed for random delay environments, it is also applicable to fixed delay environments commonly addressed in the related works. This is because fixed delay environments can be considered a special case of random delay environments, where the probability of a specific delay is equal to one.

## 5 DELAY-AWARE ACTOR-CRITIC

In Section 4, we showed how our method compensates for random delays in a simple and direct way through the value-correction method and distributional return perspective. In this section, we apply this method to derive Delay-Aware Distributional Actor-Critic (DADAC), which is designed based on the framework the Soft Actor-Critic (Haarnoja et al., 2018) and equips it to handle random delays, as shown in Figure 3.

The distributional value correction variant of the soft Bellman operator $P_{\mathcal{D}}^{\pi} : \mathcal{Z} \to \mathcal{Z}$ with the maximum entropy can be defined as

$$
\mathcal{T}_{\mathcal{D}}^{\pi} Z\left(s_t, a_t\right) \overset{D}{=} \sum_{i=1}^{\delta_o} p_i^o \cdot \sum_{j=1}^{\delta_a} p_j^a \cdot \left(r_{t+i+j} + \gamma^{i+j} \left(Z\left(s_{t+i+j+1}, a_{t+i+j+1}\right)\right.\right.
$$
$$
\left.\left. - \alpha \log \pi\left(a_{t+i+j+1} | s_{t+i+j+1}\right)\right).\right. \tag{5}
$$

As an off-policy method, our method requires more trajectory data to update once. Therefore, the proposed method samples multiple sequential trajectory information from the replay buffer concurrently, represented as $\tau_n = (s_t, a_t, r_t, s_{t+1}, a_{t+1}, \ldots, s_{\delta_o+\delta_a}, a_{\delta_o+\delta_a}, r_{\delta_o+\delta_a}, s_{\delta_o+\delta_a+1}, a_{\delta_o+\delta_a+1})$, where the length of the sampled trajectory corresponds to the maximum of total delays + 1. To implement Eq. (5), we can update the return distribution using KL divergence as loss function, which is a common strategy in the field of distributional reinforcement learning (Bellemare et al., 2017; Duan et al., 2021).

In the policy improvement step, we update the policy towards the exponential of the distributional value function. The new policy $\pi_{\text{new}}$ can be expressed as

$$
\pi_{\text{new}} = \arg \min_{\pi' \in \Pi} D_{\text{KL}} \left( \pi'(\cdot|s) \left\| \frac{\underset{Z(s,a) \sim \mathcal{Z}_{\theta}(\cdot|s,a)}{\mathbb{E}} \left[ \underset{a \sim \pi^{old}}{\mathbb{E}} \left[ Z_{\text{value}}^{\pi_{\text{old}}}(s, a) \right] \right]}{Z_{\text{action}}^{\pi_{\text{old}}}(s)} \right), \tag{6}
$$

where $Z_{\text{value}}$ denotes the value correction return distribution proposed in this paper, $\theta$ denotes the parameters of $Z_{\text{value}}$, and $Z_{\text{action}}$ denotes the action distribution. The policy can be learned by maximizing a parameterized variant of the objective as

$$J_\pi(\phi) = \mathop{\mathbb{E}}_{s\sim\mathcal{B},a\sim\pi_\phi} \left[ \mathop{\mathbb{E}}_{Z(s,a)\sim\mathcal{Z}_\theta(\cdot|s,a)} \big[Z(s,a)\big] - \alpha \log\big(\pi_\phi(a|s)\big) \right], \tag{7}$$

where $\mathcal{B}$ is the replay buffer for the information collected by the agent as it interacts with the environment, and $\phi$ denotes the parameters of policy $\pi$.

## 6 EXPERIMENTAL RESULTS

We conducted experimental evaluations in the MuJoCo environment within Gymnasium and implemented the random delay settings using Wrappers. To facilitate a comprehensive evaluation of algorithm performance, we designed two random delay distributions: a gamma distribution with mean of 2 and a double Gaussian distribution with mean of 5, the details of which is provided in Appendix. The implementation of the distributional value function in our method draws on the Distributional Soft Actor-Critic, which assumes that the random returns $Z(s,a)$ obey a Gaussian distribution (Duan et al., 2021). Given that most related work primarily considers observation delays, the following experimental results are presented in the context of observation delays for the sake of comparison.

### 6.1 COMPARATIVE EVALUATION

We compared the performance of the DADAC algorithm with the following two existing delay-aware DRL methods:

- State Augmentation-MLP is the SOTA method in non-fixed delay environments by state augmentation and recovery of delay-free trajectories (Wang et al., 2024). This method has the prior knowledge of the maximum value of the random delays.

- Belief Projection-based Q-learning (BPQL) is proposed to tackle the issue of state-space explosion caused by state augmentation in fixed delay environments through a novel projection method (Kim et al., 2023), which assumes a prior knowledge of the value of fixed delay. In the experiments (i.e. environment with random delays), for fair comparison, we use the expectation of the delay distribution as the fixed delay known to this method.

For each experiment, we performed eight runs. The results of the experiments in gamma delay distribution and double Gaussian delay distribution are shown in Figure 4 and 5, respectively. Compared to the other two methods, our proposed DADAC demonstrates significant performance advantages in the majority of experiments. In particular, the combination of the delay-aware distributional value function and value correction enables DADAC to achieve not only superior convergence speed but also reduced performance variance across multiple runs under the same conditions, showcasing strong adaptability and stability in random delay environments. The other two SOTA methods, which are impacted by random delays and face challenges in accurately representing oracle states based on their design, exhibit significant performance gaps compared to DADAC. Notably, BPQL is almost unable to learn effective policies in certain scenarios.

### 6.2 ABLATION STUDY

To better evaluate the impact of each component of our proposed algorithm, we designed the following ablation experiments involving the participating algorithms:

- Normal SAC
- SAC+Value Correction. We applied the value correction method to the normal SAC while maintaining the return as an expectation rather than as a distribution.
- Distributional Soft Actor-Critic (DSAC). A distributional SAC implementation that assumes returns as obeying a Gaussian distribution, proposed by Duan et al. (2021), without involving value correction.

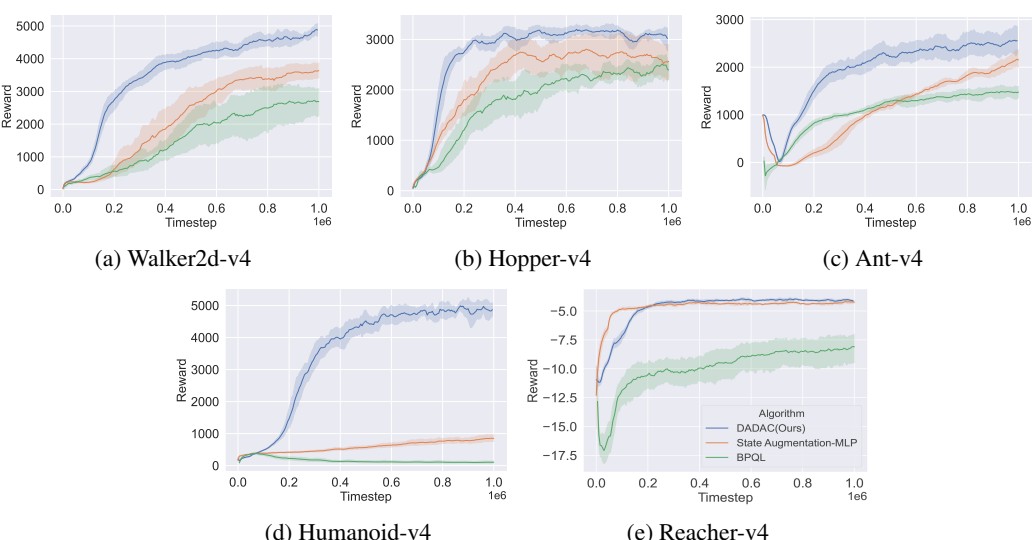

Figure 4: Comparison results in the gamma delayed environment.

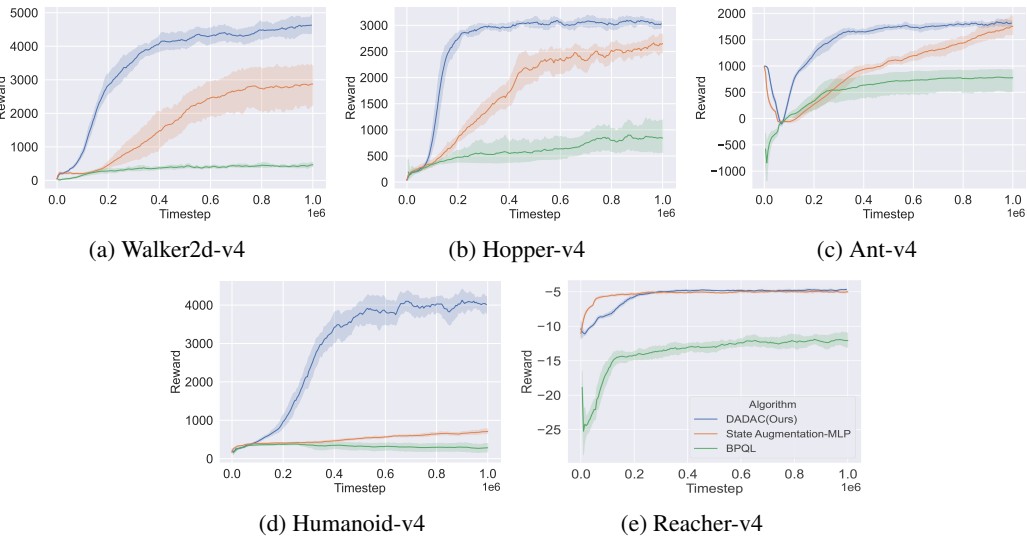

Figure 5: Comparison results in the double Gaussian delayed environment.

For each experiment, we performed eight runs, and the results of the experiments in gamma delay distribution and double Gaussian delay distribution are shown in Figure 6 and 7, respectively. Normal SAC fails to learn effective policies across nearly all scenarios due to the ineffectiveness of MDP in random delay environments, resulting in consistently the lowest rewards. Both SAC+Value Correction and the DSAC algorithm show adaptability to the delayed environment across different scenarios, offering significant performance advantages over normal SAC. However, they still experience notable performance degradation compared to our proposed DADAC, highlighting the indispensable role of value correction and the distributional value function in DADAC. It is noteworthy that both SAC+Value Correction and DSAC exhibit suboptimal performance in specific scenarios, such as Ant-v4 and Humanoid-v4, which demonstrate heightened sensitivity to delays, leading to significantly reduced rewards. In contrast, DADAC achieves markedly superior performance in these same scenarios, effectively alleviating the adverse effects of delays. This finding underscores the importance of integrating value correction mechanisms with the distributional value function, which is vital for DADAC's success and enhances its ability to navigate the complexities inherent in random delay environments more proficiently than its counterparts.

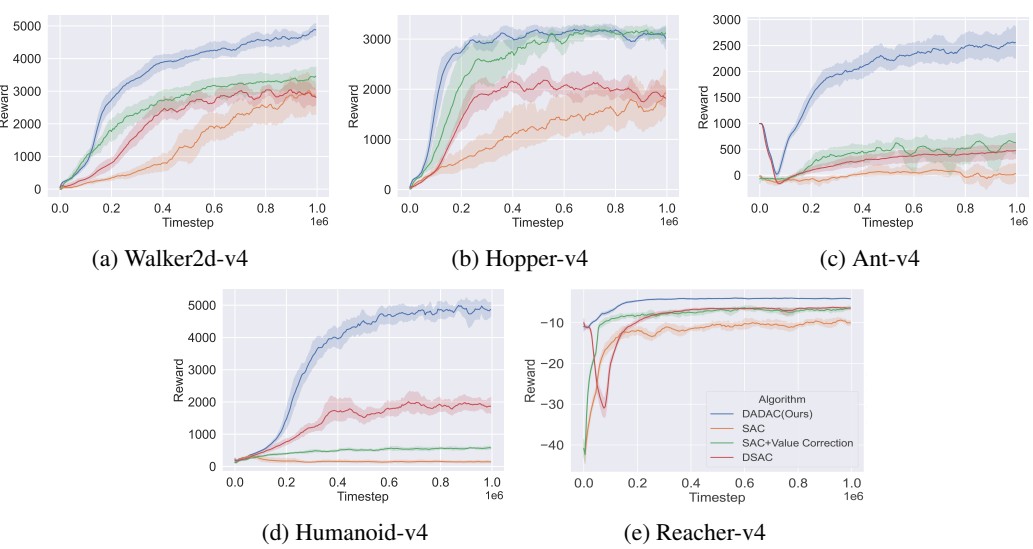

Figure 6: Ablation results in the gamma delayed environment.

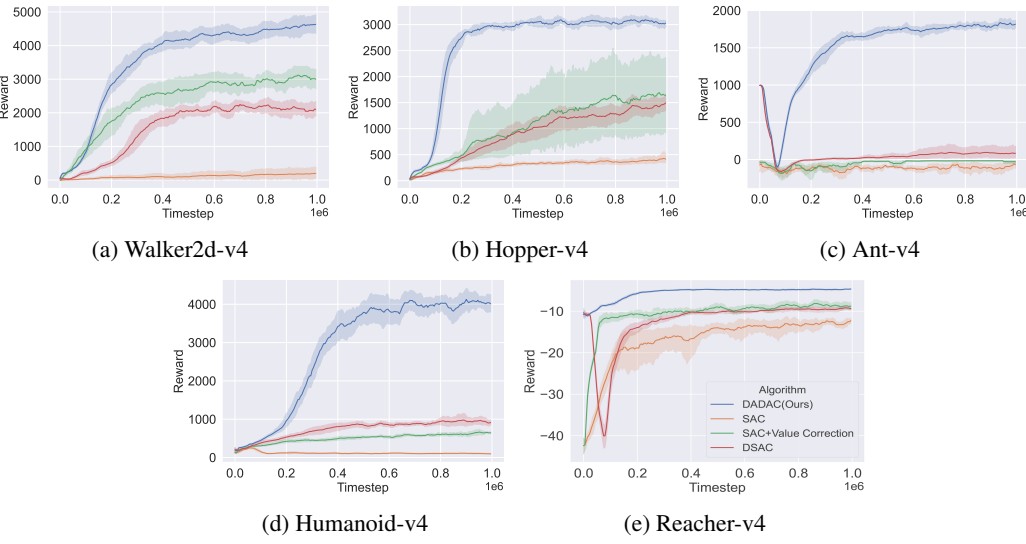

Figure 7: Ablation results in the double Gaussian delayed environment.

# 7 CONCLUSIONS

In this paper, we have tackled the challenge of random delays in the environment. To effectively model the uncertainty associated with these delays, we proposed a novel method that represents the Q value as a distribution. Building on delay distributions, we developed a value correction method to accurately recover the true return in random delay environments. By integrating these methods into the actor-critic framework, we introduced the delay-aware distributional actor-critic (DADAC) DRL method, which provides a new perspective for the field of delay-aware reinforcement learning. Our experimental results demonstrate that DADAC not only significantly outperforms the state-of-the-art delay-aware DRL methods, but also offers a robust solution to enhance performance in environments characterized by random delays.

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

# A THEOREM PROOF

## A.1 THEOREM 1 DERIVATION

*Proof.* Previous researches has proved that the distributional Bellman operator derived from the distributional Bellman equation is a contraction in terms of some measure (Bellemare et al., 2017; Nam et al., 2021).The distributional Bellman operator $\mathcal{T}^\pi : \mathcal{Z} \to \mathcal{Z}$ is defined as

$$\mathcal{T}^\pi Z(s,a) \overset{D}{:=} r(s,a) + \gamma P^\pi Z(s,a).$$

where $P^\pi : \mathcal{Z} \to \mathcal{Z}$ is a state transition operator under policy $\pi$, $P^\pi Z(s,a) \overset{D}{:=} Z(S',A')$, where $S' \sim P(\cdot|s,a)$ and $A' \sim \pi(\cdot|S')$.

Similarly, the distributional value correction variant of Bellman operator derived from the Eq. (4) can be defined as

$$\mathcal{T}^\pi Z(s,a) \overset{D}{:=} \sum_{i=1}^{\delta_o} p_i^o \cdot \sum_{j=1}^{\delta_a} p_j^a \cdot (r(s,a,s_{i+j}) + \gamma^{i+j} P^\pi Z(s_{i+j+1}, a_{i+j+1}))$$

where the $(s_{i+j}, a_{i+j})$ denotes a state-action pair that is $i + j$ timesteps later in the sequence with respect to $(s,a)$.

In this way, the distributional value correction Bellman operator can be considered equivalent to the weighted cumulative sum of several distributional Bellman operators at different times. Therefore, the distributional value correction Bellman operator is a contraction in terms of the same measure condition as the distributional Bellman operator and the value correction iteration process will converge to a unique fixed point as $t \to \infty$.

It is worth mentioning that Eq. (4) is applicable to environments where both observation and action delays are present and can be simplified to Eq. (3) when only one type of delay is present in the environment. Thus, Eq. (3) exhibits the same convergence.

## A.2 THEOREM 2 DERIVATION

*Proof.* In real-world environments, it is typically assumed that the different variants of delays are independent of each other. Consequently, for environments where both observation and action delays are present, Eq. (4) for value correction can be formulated as follows

$$Z(s_t, a_t) = \sum_{i=1}^{\delta_o} p_i^o \cdot \sum_{j=1}^{\delta_a} p_j^a \cdot (r_{t+i+j} + \gamma^{i+j} Z(s_{t+i+j+1}, a_{t+i+j+1}))$$

$$= \sum_{(i+j)=2}^{\delta_o + \delta_a} p_{i+j} \cdot (r_{t+(i+j)} + \gamma^{i+j} Z(s_{t+(i+j)+1}, a_{t+(i+j)+1}))$$

where $p_{i+j}$ denotes the probability that the delay is equal to $i + j$. From the above, it can be observed that the different variants of delays are equivalent in terms of reinforcement learning value misassignment. Consequently, the value correction method addresses variants of delays in an equivalent manner. Therefore, we can simplify the model by treating variants of delays as a single type of delay, represented by the sum of the action delay $\Delta T_a$ and the observation delay $\Delta T_o$, such that $\Delta T = \Delta T_a + \Delta T_o$, along with its distribution $d_{\Delta T_a + \Delta T_o}$.

## B EXPERIMENTAL SETUP

### B.1 RANDOM DELAY ENVIRONMENTS

To better evaluate the algorithm's performance, we design two delay distributions to simulate random delays in real application scenarios, as shown in Figure 8. The gamma delay distribution has a range of 1 to 6 with an expectation of 2, while the double Gaussian delay distribution has a range of 1 to 10 with an expectation of 5.

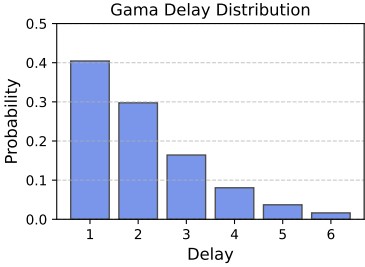   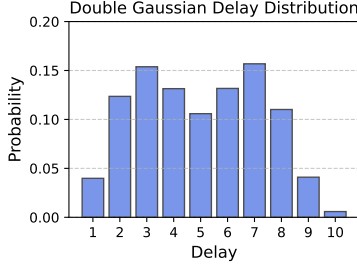

(a) Gamma Delay Distribution.          (b) Double Gaussian Delay Distribution.

Figure 8: The different distributions of random delays.

## B.2 HYPERPARAMETERS

Table 1: Hyperparameter Settings

| Hyperparameter | Setting |
|---|---|
| Network | [256, 256, 256] |
| Batch Size | 256 |
| Total Timesteps | 1000000 |
| Learning Rate | 0.0001 |
| Learning Rate for $\alpha$ | 0.0003 |
| Hidden Activation | GELU |
| Output Activation | Linear |
| $\gamma$ | 0.99 |
| Optimizer | Adam |
| Initial $\alpha$ | 0.2 |

