# OpenReview forum: "Delay-Aware Reinforcement Learning: Insights From Delay Distributional Perspective"
_ICLR.cc/2025/Conference — Submitted to ICLR 2025_

### Official Review · Reviewer_HieD · 2024-10-21

**Soundness:** 3
**Presentation:** 1
**Contribution:** 2
**Rating:** 5
**Confidence:** 5

**Summary:**

This paper proposes a distributional delayed RL method named DADAC from the perspective of distributional RL. The theoretical result shows that the proposed Distributional Value Correction Bellman Equation has the convergence guarantee. The experimental results show that the

**Strengths:**

1. The motivation behind the proposed method is straightforward.
2. The theoretical analysis of the convergence makes sense.

**Weaknesses:**

The proposed method's motivation is straightforward. However, this paper ignores too many closely related references and/or important baselines. Therefore, the reviewer must point out these issues.

## Writing
1. The description should be unified across the paper. For instance, times $t+1$ (lines 232), $i$ timesteps (lines 239), $i$-th step (lines 240), and $(t+i+j)$-th time step (lines 250).
2. There are multiple different definitions of the reward function. For instance, $r(s,a)$ in Eq.(1), $r(s,a,s')$ in Eq.(2), $r_{t+i}(s_t, a_t, s_{t+i})$ in Eq.(3). Especially, what's the meaning of $r_{t+i}(s_t, a_t, s_{t+i})$.
3. Typo issues. $i$-th step (lines 240). $(t+i+j)$-th time step (lines 250).

## Theorem
1. The theorem 1 just shows the contraction property based on the existing literature, but the reviewer looks forward to the authors giving the analysis on the fixed point which can measure the performance of the proposed operator in the delayed RL settings.
2. The equivalent of different delays (theorem 2) has been proven by previous works [1, 2] experimentally and theoretically. The authors should highlight the difference with previous works if providing novel theoretical contributions.

## Experiment
1. The performance of BPQL[8] and State-Aug-MLP[9] is doubtful. The setting for the constant delayed RL methods is not fair somehow (using the expectation of the delay distribution). From the perspective of the reviewer, using the maximum delay is more fair. Specifically, based on the experimental results provided in the BPQL paper, the performance in Walker2d-v3 with the 9 constant delays is 4104.3, showing a serious performance drop (around 2700.0 in Walker2d-v4) in this paper (Figure 4).
2. DATS[5], DIDA[6], and AD-RL[7] should be considered as baselines in the constant delay settings.
3. RTAC[3], DC/AC[4] and VDPO[10] should be considered as baselines in the stochastic delay settings. In particular, VDPO[10] can be regarded as a recent work, the performance comparison is not compulsory.


[1] Katsikopoulos, Konstantinos V., and Sascha E. Engelbrecht. "Markov decision processes with delays and asynchronous cost collection." IEEE transactions on automatic control 48.4 (2003): 568-574.

[2] Nath, Somjit, Mayank Baranwal, and Harshad Khadilkar. "Revisiting state augmentation methods for reinforcement learning with stochastic delays." Proceedings of the 30th ACM International Conference on Information & Knowledge Management. 2021.

[3] Simon Ramstedt, et al. Real-time reinforcement learning. In NeurIPS, 2019.

[4] Bouteiller, Yann, et al. "Reinforcement learning with random delays." International conference on learning representations. 2020.

[5] Chen, Baiming, et al. "Delay-aware model-based reinforcement learning for continuous control." Neurocomputing 450 (2021): 119-128.

[6] Liotet, Pierre, et al. "Delayed reinforcement learning by imitation." International Conference on Machine Learning. PMLR, 2022.

[7] Wu, Qingyuan, et al. "Boosting Reinforcement Learning with Strongly Delayed Feedback Through Auxiliary Short Delays." Forty-first International Conference on Machine Learning.

[8] Kim, Jangwon, et al. "Belief projection-based reinforcement learning for environments with delayed feedback." Advances in Neural Information Processing Systems 36 (2023): 678-696.

[9] Wang, Wei, et al. "Addressing Signal Delay in Deep Reinforcement Learning." The Twelfth International Conference on Learning Representations. 2023.

[10] Wu, Qingyuan, et al. "Variational Delayed Policy Optimization." Advances in Neural Information Processing Systems 37 (2024).

**Questions:**

See Weakness.

---

> ### Author Response · Authors · 2024-11-28
>
> We greatly appreciate your thoughtful and detailed review of our work. The following responses address your concerns in detail.
>
> - [W1] Writing
>
> We sincerely appreciate your thorough reading and careful review of our paper. In future revisions, we will address and resolve the issues you raised to ensure a more polished and unified presentation.
>
> - [W2] Theorem
>
> For Theorem 1, we will investigate the performance of the proposed operator at the fixed point in the future version of this paper. For Theorem 2, we focused on validating the applicability of the equivalence between different types of delays within the framework of our method, where delays are assumed to follow a specific probability distribution.
>
> - [W3] Experiment
>
> **1. Doubts about the performance of BPQL[5] and State-Aug-MLP[6]**
>
> Experiments with BPQL[5] in gamma delay distribution at different delay settings
> Environment     | BPQL (using the mean of 2) |  BPQL(using the maximum of 6) |
> -------- | -------- | --------- |
> Walker2d-v4  | **2450.8 ± 1088.5** | -5.5 ± 4.8 |
> Hopper-v4  | **2408.0 ± 459.3** |  7.9 ± 5.8 |
> Ant-v4  | **1552.1 ± 237.2** | 111.1 ± 328.7 |
> Humanoid-v4 | **99.0 ± 30.2** | 82.9 ± 12.6 |
> Reacher-v4 | **-8.5 ± 2.2** | -32.8 ± 8.5
>
> Experiments with BPQL[5] in double gaussian delay distribution at different delay settings
> Environment     | BPQL (using the mean of 5) |  BPQL(using the maximum of 10) |
> -------- | -------- | --------- |
> Walker2d-v4  | **517.3 ± 88.3** | 1.5 ± 7.2 |
> Hopper-v4  | **752.5 ± 465.9** | 11.4 ± 9.1  |
> Ant-v4  | **791.2 ± 325.5** | 545.2 ± 286.3 |
> Humanoid-v4 | **272.2 ± 162.6** | 114.6 ± 16.7 |
> Reacher-v4 | **-11.7 ± 1.4** | -28.4 ± 10.8 |
>
> In random delay environments, we provide the mean of the delay distribution rather than its maximum value to BPQL[5], as BPQL fundamentally relies on a state augmentation approach. We believe that the augmented state represents a potential expression of the ideal, delay-free observation. For example, if we were to provide BPQL with the maximum delay value—such as the gamma distribution with a range of 1-6 (mean 2) used in this paper—BPQL would always select actions based on the state 6 time steps ahead. However, in reality, the probability of a delay of 6 is extremely low. This setup would likely make the algorithm overly farsighted. Providing BPQL with the mean delay instead allows it to consistently select actions in an averaged manner. To validate this, we added experiments where the maximum delay value was provided to BPQL. As expected, this configuration further degraded BPQL's performance.
>
> For the State-Aug-MLP[6] method, the original paper explicitly specifies the use of the maximum value of random delays in non-fixed delay environments. Therefore, we followed this setup and provided the maximum random delay value to State-Aug-MLP.
>
> Regarding the significant performance decline of BPQL in our experiments, **we believe this is inherently caused by the differences between fixed-delay and random-delay environments**. In random-delay environments, observations or actions can experience time misalignment, repetition, or omission. These factors significantly impair the ability of state augmentation methods to accurately represent the ideal delay-free observation, further disrupting the Markov property and severely affecting the algorithm's performance. This highlights the primary challenge in addressing delay issues in random-delay environments. Lastly, we assure that the experimental data presented in our results are genuine and accurate, and we have released the complete source code of our method.
>
> **2&3. Additional baselines**
>
> Among the additional baselines mentioned by the reviewers, DC/AC[1] is specifically designed for addressing random delay. However, due to its computational inefficiency, we decided not to include comparisons with it, and its limitation is also discussed in [2]. Based on our investigation, other baselines are primarily designed for fixed-delay environments, although some methods briefly evaluate their performance in random-delay environments.
>
> For instance, RTAC[7] is designed to handle one-step action delays, which makes it less suitable for comparison with delay-aware RL methods. Additionally, DIDA[2] has not released its source code. Due to time constraints, we conducted limited tests to include comparisons with VDPO[4], which is likely one of the most recent methods. Notably, VDPO outperforms AD-RL[3], DIDA[2], DC/AC[1] in most scenarios according to its reported experimental results.

---

> ### Author Response · Authors · 2024-11-28
>
> **The following are the latest comparison evaluation results.**
>
> Comparative evaluation in gamma delay distribution
> Environment     | BPQL[5] | State Augmentation-MLP[6] | VDPO[4] | DADAC(Ours)
> -------- | -------- | --------- | -------- | -------- |
> Walker2d-v4  | 2450.8 ± 1088.5 | 3761.0 ± 278.6 | 3300.9 ± 976.1 | **4928.5 ± 461.9** |
> Hopper-v4  | 2408.0±459.3 | **2788.2 ± 690.7** | 1469.9 ± 214.1 | 2696.3 ± 810.0 |
> Ant-v4  | 1552.1±237.2 | 1996.6 ± 488.0 | 1453.5 ± 474.4 | **2508.5 ± 492.5** |
> Humanoid-v4 | 99.0±30.2 | 899.4 ± 204.6 | 2469.9 ± 2160.8 | **5161.9 ± 490.9** |
> Reacher-v4 | -8.5 ± 2.2 | -4.6 ± 0.4 | -5.5 ± 1.4 | **-4.1 ± 0.5** |
>
>
>
> Comparative evaluation in double gaussian delay distribution
> Environment     | BPQL[5] | State Augmentation-MLP[6] | VDPO[4] | DADAC(Ours)
> -------- | -------- | --------- | -------- | -------- |
> Walker2d-v4  | 517.3 ± 88.3 | 2624.9 ± 911.5 | 1098.1 ± 810.0 | **4693.4 ± 792.1** |
> Hopper-v4  | 752.5 ± 465.9 | 2731.1 ± 446.0 | 611.2 ± 286.0 | **3091.1 ± 284.0** |
> Ant-v4  | 791.2 ± 325.5 | **1788.8 ± 348.0** | 1365.1 ± 531.3 | 1778.8 ± 105.3 |
> Humanoid-v4 | 272.2 ± 162.6 | 714.2 ± 123.2 | 1938.0 ± 122.0 | **4023.9 ± 722.6** |
> Reacher-v4 | -11.7 ± 1.4 | -4.8 ± 1.1 | -7.2 ± 1.0 | **-4.8 ± 0.3** |
>
>
>
> Comparative evaluation in uniform delay distribution
> Environment     | BPQL[5] | State Augmentation-MLP[6] | VDPO[4] | DADAC(Ours)
> -------- | -------- | --------- | -------- | -------- |
> Walker2d-v4  | 403.4 ± 116.0 | 1426.8 ± 532.0 | 901.1 ± 222.0 | **4764.6 ± 388.1** |
> Hopper-v4  | 372.7 ± 20.6 | 2442.9 ± 318.0 | 433.3 ± 27.5 | **3116.0 ± 112.7** |
> Ant-v4  | 498.5 ± 314.3 | 1829.7 ± 99.7 | 1282.0 ± 211.1 | **2077.4 ± 44.3** |
> Humanoid-v4 | 146.2 ± 51.4 | 734.6 ± 71.0 | 3622.2 ± 1418.4 | **4195.5 ± 464.9** |
> Reacher-v4 | -13.8 ± 1.8 | **-4.5 ± 0.9** | -10.3 ± 2.4 | -5.4 ± 0.4 |
>
> References
>
> [1]Bouteiller, Y., Ramstedt, S., Beltrame, G., Pal, C., and Binas, J. Reinforcement learning with random delays. In International conference on learning representations, 2021.
>
> [2]Liotet, P., Maran, D., Bisi, L., and Restelli, M. Delayed reinforcement learning by imitation. In International Conference on Machine Learning, pp. 13528–13556. PMLR, 2022.
>
> [3]Qingyuan Wu, Simon Sinong Zhan, Yixuan Wang, Yuhui Wang, Chung-Wei Lin, Chen Lv, Qi Zhu, Jürgen Schmidhuber, and Chao Huang. Boosting reinforcement learning with strongly delayed feedback through auxiliary short delays. In Forty-first International Conference on Machine Learning(2024).
>
> [4]Qingyuan Wu, Simon Sinong Zhan, Yixuan Wang, Yuhui Wang, Chung-Wei Lin, Chen Lv, Qi Zhu, and Chao Huang. Variational delayed policy optimization. In Advances in Neural Information Processing Systems 37 (2024).
>
> [5]Jangwon Kim, Hangyeol Kim, Jiwook Kang, Jongchan Baek, and Soohee Han. Belief projection based reinforcement learning for environments with delayed feedback. Advances in Neural Information Processing Systems, 36:678–696, 2023.
>
> [6]Wei Wang, Dongqi Han, Xufang Luo, and Dongsheng Li. Addressing signal delay in deep reinforcement learning. In The Twelfth International Conference on Learning Representations, 2024.
>
> [7] Simon Ramstedt, et al. Real-time reinforcement learning. In NeurIPS, 2019.

---

### Official Review · Reviewer_4xeB · 2024-10-22

**Soundness:** 2
**Presentation:** 2
**Contribution:** 3
**Rating:** 5
**Confidence:** 4

**Summary:**

This paper proposes a novel approach to solve the random delay problem in the delayed reinforcement learning by using distributional RL techniques to capture both state and action delays in the observation signals.

**Strengths:**

Author did a great job illustrating the motivation of the work, making it intuitive and easy to understand. Most parts of the paper is well-written and structured with logic coherence.

**Weaknesses:**

### Related Works
1, Missing some recent benchmarks: DC/AC[1], AD-RL[2], DIDA[3], VDPO[4], RTAC[5]

### Section 4
1, Assumptions should be clearly stated either in the prelim part or the start of this section.\
2, the reward defined in Eq 2, 3, and 4 are all different, which is quite misleading for interpretation. \
3, Proof of Thm 1 is not rigorous: a) Reviewer expect the author to write out a few more steps on bellman property of newly defined bellman operator for both stationary and non-stationary delay distribution if stationarity of delay distribution is not universal. Otherwise Thm1 seems not be sufficient enough as an independent theorem. \
4, Following up to the convergence, it would be more technical solid to analyze optimality of the method, since convergence itself cannot provide any info on optimality. In [2,3,4], their optimality is investigated through study on the fixed point of optimization, which could be a possible direction of extension.

### Experiments
1, The review suggests to involve more baselines for comparison aside from mentioned approaches, at least the recent works DIDA, AD-RL, DC/AC, etc. Current experiments seem not be sufficient enough to support author's arguments.\
2, All the experiments are conducted in the deterministic MDP. It would be interesting to have some analysis/ablation studies on the stochastic setting, since the proposed method seems to support the stochastic MDPs.

[1]Bouteiller, Y., Ramstedt, S., Beltrame, G., Pal, C., and Binas, J. Reinforcement learning with random delays. In International conference on learning representations, 2020.\
[2]Qingyuan Wu, Simon Sinong Zhan, Yixuan Wang, Yuhui Wang, Chung-Wei Lin, Chen Lv, Qi Zhu, Jürgen Schmidhuber, and Chao Huang. Boosting reinforcement learning with strongly delayed feedback through auxiliary short delays. In Forty-first International Conference on Machine Learning.\
[3]Liotet, P., Maran, D., Bisi, L., and Restelli, M. Delayed reinforcement learning by imitation. In International Conference on Machine Learning, pp. 13528–13556. PMLR, 2022.\
[4]Qingyuan Wu, Simon Sinong Zhan, Yixuan Wang, Yuhui Wang, Chung-Wei Lin, Chen Lv, Qi Zhu, and Chao Huang. Variational delayed policy optimization. In Advances in Neural Information Processing Systems 37 (2024).\
[5]Simon Ramstedt, et al. Real-Time Reinforcement Learning. In Advances in Neural Information Processing Systems 32 (2019).\
[6]Bellemare, Marc G., Will Dabney, and Rémi Munos. "A distributional perspective on reinforcement learning." International conference on machine learning. PMLR, 2017.

**Questions:**

1, Is the delay distribution a required prior knowledge and also stationary? If so, what is the rationale picking gamma and double gaussian? What if an uniform distribution? \
2, How the prior delay distribution will affect the final return distribution of the DRL in both theoretical and empirical way, since no matter what the return of DRL seems to be assumed as a Gaussian? Only mean and log_std of Gaussian are affected or the way to choose value instead of expectation? \
3, Out of curiosity, the observation-delayed MDP have been proved to be the superset of the action-delayed MDP problem[7,8] also stated in Thm 2. Why not simplify the Eq (4) where both delay are considered to some form of Eq (3) for simplicity of unification?


[7]Katsikopoulos, K. V. and Engelbrecht, S. E. Markov decision processes with delays and asynchronous cost collection. IEEE transactions on automatic control, 48(4):568–574, 2003.\
[8]Nath, S., Baranwal, M., and Khadilkar, H. Revisiting state augmentation methods for reinforcement learning with stochastic delays. In Proceedings of the 30th ACM International Conference on Information & Knowledge Management, pp. 1346–1355, 2021.

---

> ### Author Response · Authors · 2024-11-28
>
> We sincerely thank you for your meticulous review and valuable suggestions. The following are our responses to your comments.
>
> - [Q1] Concerns about delay distribution
>
> Our method assumes that the delay distribution is known as prior knowledge and that it possesses stationary properties. In recent related research, most methods typically rely on exact delay information as prior knowledge, such as real-time true delay values [1,2,3,4,5,7,8], and most of them are designed for the environment with fixed delays. However, this type of prior knowledge is often difficult to obtain in practical applications and the delays are usually non-fixed in practical systems. This study aims to bring delay-aware reinforcement learning in random delay environments closer to real-world applications.
>
> Some existing work also assumes prior knowledge of the maximum value of random delays [6]，but we found that many real-world delays often follow specific statistical patterns [6,9,10,12], such as network delays. By leveraging these statistical characteristics as prior knowledge for the delay distribution, our method becomes more suitable for real-world scenarios and outperforms the method with prior knowledge of maximum delay. The analysis of Wi-Fi network delays in [1] shows that the statistical results closely resemble a gamma distribution; in [9], the authors used Gaussian-related distributions to simulate delays. These studies inspired us to design gamma and double Gaussian distributions to simulate various delay environments.
>
> Theoretically, as long as the delay distribution is known, our method can effectively address delay problems, regardless of the specific type of delay distribution. To further validate the adaptability of our method across different delay environments, we have included experimental results based on a uniform distribution with values ranging from 1 to 13.
>
> - [Q2] How the delay distribution affects the return distribution?
>
> Distributional reinforcement learning aims to more accurately represent the return in the form of a distribution, and we refer to [11] and estimate the return using a Gaussian distribution. In our method, the critic learns the mean and standard deviation of this return distribution to represent the final return of a state-action pair, accounting for the impact of the delay distribution. During the policy improvement phase, we use the mean of the return distribution to assist in policy updates.
>
> - [Q3] Why Eq. (4) is not reduced to a unified form.
>
> As the reviewer pointed out, existing work has demonstrated the equivalence of observation delay and action delay. In Theorem 2, we also prove that our method can equivalently handle both types of delays. In the presentation of this paper, we extended the consideration from only one type of delay to simultaneously considering both types. In fact, equation (3) is a simplified version of equation (4), which only considers one type of delay, as long as we assume that $p_i$ represents a probability distribution equivalent to the cumulative effect of both types of delays. However, in practical applications, if the delay distributions for observation and action delays are different, such as the observation delay following a gamma distribution and the action delay following a double Gaussian distribution, the probability distribution equivalent to the cumulative effect of these delay distributions is difficult to compute accurately. Therefore, we retain the formulation that simultaneously considers both types of delays for ease of understanding and calculation.
>
> - [W1] Related Works
>
> We will include the missing recent benchmarks in future versions of the paper.
>
> - [W2] Section 4
>
> We sincerely thank the reviewers for their writing suggestions and for pointing out the issues with inconsistent definitions. We will carefully consider the reviewers' concerns regarding the proof of Theorem 1 and potential extensions, and we will make the necessary revisions in future versions of the paper.
>
> - [W3] Experiments
>
> Due to time constraints, we only added comparisons with VDPO [4], which is one of the most recent methods, and VDPO [4] outperforms all the other methods including DIDA[2], AD-RL[3], DC/AC[1] in most scenarios. (Additionally, DIDA did not released its source code. We initially considered comparing with DC/AC[1]; however, its computational inefficiency led us to abandon this comparison, and its limitation is also discussed in [2].) In future versions of the paper, we plan to further refine the experimental results and incorporate additional baselines.

---

> ### Author Response · Authors · 2024-11-28
>
> **The following are the latest comparison evaluation results.**
>
> Comparative evaluation in gamma delay distribution
> Environment     | BPQL[5] | State Augmentation-MLP[6] | VDPO[4] | DADAC(Ours)
> -------- | -------- | --------- | -------- | -------- |
> Walker2d-v4  | 2450.8 ± 1088.5 | 3761.0 ± 278.6 | 3300.9 ± 976.1 | **4928.5 ± 461.9** |
> Hopper-v4  | 2408.0±459.3 | **2788.2 ± 690.7** | 1469.9 ± 214.1 | 2696.3 ± 810.0 |
> Ant-v4  | 1552.1±237.2 | 1996.6 ± 488.0 | 1453.5 ± 474.4 | **2508.5 ± 492.5** |
> Humanoid-v4 | 99.0±30.2 | 899.4 ± 204.6 | 2469.9 ± 2160.8 | **5161.9 ± 490.9** |
> Reacher-v4 | -8.5 ± 2.2 | -4.6 ± 0.4 | -5.5 ± 1.4 | **-4.1 ± 0.5** |
>
>
>
> Comparative evaluation in double gaussian delay distribution
> Environment     | BPQL[5] | State Augmentation-MLP[6] | VDPO[4] | DADAC(Ours)
> -------- | -------- | --------- | -------- | -------- |
> Walker2d-v4  | 517.3 ± 88.3 | 2624.9 ± 911.5 | 1098.1 ± 810.0 | **4693.4 ± 792.1** |
> Hopper-v4  | 752.5 ± 465.9 | 2731.1 ± 446.0 | 611.2 ± 286.0 | **3091.1 ± 284.0** |
> Ant-v4  | 791.2 ± 325.5 | **1788.8 ± 348.0** | 1365.1 ± 531.3 | 1778.8 ± 105.3 |
> Humanoid-v4 | 272.2 ± 162.6 | 714.2 ± 123.2 | 1938.0 ± 122.0 | **4023.9 ± 722.6** |
> Reacher-v4 | -11.7 ± 1.4 | -4.8 ± 1.1 | -7.2 ± 1.0 | **-4.8 ± 0.3** |
>
>
>
> Comparative evaluation in uniform delay distribution
> Environment     | BPQL[5] | State Augmentation-MLP[6] | VDPO[4] | DADAC(Ours)
> -------- | -------- | --------- | -------- | -------- |
> Walker2d-v4  | 403.4 ± 116.0 | 1426.8 ± 532.0 | 901.1 ± 222.0 | **4764.6 ± 388.1** |
> Hopper-v4  | 372.7 ± 20.6 | 2442.9 ± 318.0 | 433.3 ± 27.5 | **3116.0 ± 112.7** |
> Ant-v4  | 498.5 ± 314.3 | 1829.7 ± 99.7 | 1282.0 ± 211.1 | **2077.4 ± 44.3** |
> Humanoid-v4 | 146.2 ± 51.4 | 734.6 ± 71.0 | 3622.2 ± 1418.4 | **4195.5 ± 464.9** |
> Reacher-v4 | -13.8 ± 1.8 | **-4.5 ± 0.9** | -10.3 ± 2.4 | -5.4 ± 0.4 |
>
> References
>
> [1]Bouteiller, Y., Ramstedt, S., Beltrame, G., Pal, C., and Binas, J. Reinforcement learning with random delays. In International conference on learning representations, 2021.
>
> [2]Liotet, P., Maran, D., Bisi, L., and Restelli, M. Delayed reinforcement learning by imitation. In International Conference on Machine Learning, pp. 13528–13556. PMLR, 2022.
>
> [3]Qingyuan Wu, Simon Sinong Zhan, Yixuan Wang, Yuhui Wang, Chung-Wei Lin, Chen Lv, Qi Zhu, Jürgen Schmidhuber, and Chao Huang. Boosting reinforcement learning with strongly delayed feedback through auxiliary short delays. In Forty-first International Conference on Machine Learning (2024).
>
> [4]Qingyuan Wu, Simon Sinong Zhan, Yixuan Wang, Yuhui Wang, Chung-Wei Lin, Chen Lv, Qi Zhu, and Chao Huang. Variational delayed policy optimization. In Advances in Neural Information Processing Systems 37 (2024).
>
> [5]Jangwon Kim, Hangyeol Kim, Jiwook Kang, Jongchan Baek, and Soohee Han. Belief projection based reinforcement learning for environments with delayed feedback. Advances in Neural Information Processing Systems, 36:678–696, 2023.
>
> [6]Wei Wang, Dongqi Han, Xufang Luo, and Dongsheng Li. Addressing signal delay in deep reinforcement learning. In The Twelfth International Conference on Learning Representations, 2024.
>
> [7]Baiming Chen, Mengdi Xu, Liang Li, and Ding Zhao. Delay-aware model-based reinforcement learning for continuous control. Neurocomputing, 450:119–128, 2021.
>
> [8]Somjit Nath, Mayank Baranwal, and Harshad Khadilkar. Revisiting state augmentation methods for reinforcement learning with stochastic delays. In Proceedings of the 30th ACM International Conference on Information & Knowledge Management, pp. 1346–1355, 2021.
>
> [9]Krasniqi, F., Elias, J., Leguay, J., & Redondi, A. E. End-to-end delay prediction based on traffic matrix sampling. In IEEE INFOCOM 2020-IEEE Conference on Computer Communications Workshops (INFOCOM WKSHPS) (pp. 774-779). IEEE.
>
> [10]Wang, Y., Vuran, M. C., & Goddard, S. Cross-layer analysis of the end-to-end delay distribution in wireless sensor networks. IEEE/ACM Transactions on networking, 20(1), 305-318.
>
> [11]Jingliang Duan, Yang Guan, Shengbo Eben Li, Yangang Ren, Qi Sun, and Bo Cheng. Distributional soft actor-critic: Off-policy reinforcement learning for addressing value estimation errors. IEEE transactions on neural networks and learning systems, 33(11):6584–6598, 2021.
>
> [12]Guillo-Sansano, E., Syed, M. H., Roscoe, A. J., Burt, G. M., & Coffele, F.. Characterization of time delay in power hardware in the loop setups. IEEE Transactions on Industrial Electronics, 2020, 68(3), 2703-2713.

---

> ### Comment · Reviewer_4xeB · 2024-12-01
>
> Thanks for author’s rebuttal, and some answers do address my concern. Here are some remaining concerns:
> * All the addressed writing issues etc seems not be addressed in the updated pdf.
> * Under what kind of scenarios we have both action delay and state delay with different prior distribution? And do the system still possess markovian property? If answer is yes, then I can understand the not unified version of Eq 4, but corresponding proof need to provided.
> * I think DC/AC and AD-RL as the well-known works in the field should appear as benchmarks for comparison, though VDPO seems to be the recently appeared work. It only address the sample efficiency issue in previous works with similar convergence point guarantee, and in their experiments there is also no statement or proof for exceeding final performance.
> * The problem of Thm 1 is still not resolved.
>
> Thus, I have decided to remain my current scores and confidence if no further improvements have been made.

---

### Official Review · Reviewer_VDd6 · 2024-10-25

**Soundness:** 2
**Presentation:** 3
**Contribution:** 2
**Rating:** 3
**Confidence:** 3

**Summary:**

This paper introduces an innovative approach that integrates distributional value function and value correction mechanism to handle random and unpredictable delay environments. Experiments are conducted on MuJoCo, comparing with baselines of State Augmentation-MLP and BPQL.

**Strengths:**

•	Overall, the concept presented in the paper is simple and straightforward. It is interesting to use a value correction mechanism targeting the value function.

•	The results show that DADAC generally surpasses two baselines, and achieves outstanding performance in some environments. The ablation study shows the effectiveness of both the distributional value function and value correction mechanism.

**Weaknesses:**

•	The authors mentioned that observation and action delays are considered. However, there is a lack of relevant experiments to support their claims. The absence undermines the credibility of the proposed method's robustness in scenarios where both types of delays occur simultaneously.

•	It seems that the authors do not discuss the reason why using the correction mechanism on the distribution of return, although the ablation study shows its effectiveness. Besides, this mechanism assumes precise estimation of delay dynamics, which might not be easy to obtain in practice.

•	The experiments are insufficient. Only two baseline models are compared, and the results are primarily visualized through training curves, lacking other forms of analysis. And the two random delay distributions are not sufficiently varied to capture the true random delays. Also, the algorithms employed in the ablation study can be replaced with more advanced ones.


Minor comments:

•	There is a typo "a gama distribution", it should be "a gamma distribution".

•	The presentation of related work could be more concise. Specifically, the discussion of prior methods could be less detailed in the introduction and moved to the related work.

**Questions:**

1.	Why focus on correcting the distribution of return rather than simply correcting the expected value of return?

2.	How do you calculate $p_i$ in Equation (3)? If $p_i$ is constant, Equation (3) is essentially the same as the Bellman equation.

3.	In Experimental Results, how do you determine the mean of gamma distribution and double Gaussian distribution? Why do you use these two distributions? Can you use a different distribution with a random mean to re-implement the experiment?

4.	Considering fixed delays are still common in the real world, can you implement DADAC under fixed delays?

5.	How does DADAC compare to a delay-free algorithm like robust RL?

---

> ### Author Response · Authors · 2024-12-03
>
> Thank you for your detailed review and valuable insights. Below please find our response to your concerns.
>
> - [Q1 & W2] Concerns about value correction mechanism
>
> Firstly, in our method, distributional returns and the value correction mechanism are complementary and interdependent. Random delays may introduce issues such as missing, duplicate, or out-of-order observations and actions, resulting in sequential feedback from the agent-environment interaction that is inconsistent with the feedback stored in the replay buffer. Due to this inconsistency, SAC+Value Correction suffers from inaccurate Q-value estimation based on sampled data from the replay buffer, thereby reducing the effectiveness of the value correction mechanism.
>
> In contrast, distributional returns can effectively model the uncertainty in delayed environments, enhancing the robustness and multimodal expressiveness of the value function estimation. Consequently, the DSAC method inherently demonstrates a certain level of adaptability to delayed environments, achieving performance comparable to SAC+Value Correction. Furthermore, leveraging the advantage of distributional returns in significantly reducing value function estimation errors, the value correction mechanism becomes more effective. This synergy enables our proposed DADAC method to achieve superior performance in random delay environments.
>
> - [Q2 & Q4] Concerns about the settings of delay distribution
>
> In equation (3), $p_i$ represents the probability of the delay of i, which is typically considered a fixed decimal value between 0 and 1. This delay distribution acts as prior knowledge for the value correction mechanism, with the assumption that the delay distribution is stationary. If $p_i$ = 1, it indicates a constant delay environment with the delay of i. In this case, equation (3) can be written as
>
> $Z(s_{t},a_{t})=r_{t+i}(s_{t},a_{t},s_{t+i})+\gamma^{i}\cdot\sum_{a_{t+i}\sim\pi(\cdot|s_{t+i})}Z(s_{t+i},a_{t+i})$
>
> This is not the conventional Bellman equation, as the right-hand side represents the return starting from time t+i. The variant form of equation (3) is precisely how our method handles fixed delay environments. The following are the experimental results of our method in constant delay environment with limited tests. The results demonstrate that DADAC effectively handles constant delay environments.
>
> Comparative evaluation with constant delay of 5
> Environment     | BPQL | State Augmentation-MLP | VDPO | DADAC(Ours)
> -------- | -------- | --------- | -------- | -------- |
> Walker2d-v4  |  4369.0 ± 538.9 | 3917.0 ± 190.8 | 3017.4 ± 890.5 | **4475.1 ± 501.8** |
> Hopper-v4  | **3164.8 ± 449.1** | 2624.2 ± 545.6 | 2509.4 ± 727.6 | 2694.6 ± 484.0 |
> Ant-v4  | **5433.8 ± 28.7** | 2529.5 ± 998.7 | 4868.6 ± 832.2 |  2007.1 ± 54.2 |
> Humanoid-v4 |  **5371.4 ± 72.2** | 586.2 ± 87.1 | 4938.0 ± 670.5 | 4514.6 ± 579.8 |
> Reacher-v4 | -6.1 ± 1.5 | **-4.9 ± 0.4** | -5.8 ± 1.3 | -5.3 ± 0.2 |
>
> - [Q3 & W2] Questions about delay distributions in experiments
>
> Our method assumes that the delay distribution is known as prior knowledge and that it possesses stationary properties. In recent related research, most methods typically rely on exact delay information as prior knowledge, such as real-time true delay values [1,2,3,4,5,7,8], and most of them are designed for the enviornment with fixed delays. However, this type of prior knowledge is often difficult to obtain in practical applications and the delays are usually non-fixed in practical systems. This study aims to bring delay-aware reinforcement learning in random delay environments closer to real-world applications.
>
> Some existing work also assumes prior knowledge of the maximum value of random delays [6]，but we found that many real-world delays often follow specific statistical patterns [6,9,10,11], such as network delays. By  leveraging these statistical characteristics as prior knowledge for the delay distribution, our method becomes more suitable for real-world scenarios and outperforms the method with prior knowledge of maximum delay. The analysis of Wi-Fi network delays in [1] shows that the statistical results closely resemble a gamma distribution; in [9], the authors used Gaussian-related distributions to simulate delays. These studies inspired us to design gamma and double Gaussian distributions to simulate various delay environments.
>
> Theoretically, as long as the delay distribution is known, our method can effectively address delay problems, regardless of the specific type of delay distribution. To further validate the adaptability of our method across different delay environments, we have included experimental results based on a uniform distribution with values ranging from 1 to 13.

---

> ### Author Response · Authors · 2024-12-03
>
> - [W3] The experiments are insufficient
>
> The two methods we compared, BPQL[5] and State Augmentation-MLP[6], are already state-of-the-art methods. While we could compare with other methods, their performance is inferior to these two SOTA baselines. Additionally, we identified a newly published method, VDPO[4], which represents the latest work in this area. To further enrich our experiments, we added comparisons with VDPO[4] and included environments where the delay distribution follows a uniform distribution ranging from 1 to 13. Due to time constraints, we conducted a limited set of experimental tests. In future work, we plan to refine the experimental results further and consider incorporating more baselines and diverse delay environments.
>
> Comparative evaluation in gamma delay distribution
> Environment     | BPQL | State Augmentation-MLP | VDPO | DADAC(Ours)
> -------- | -------- | --------- | -------- | -------- |
> Walker2d-v4  | 2450.8 ± 1088.5 | 3761.0 ± 278.6 | 3300.9 ± 976.1 | **4928.5 ± 461.9** |
> Hopper-v4  | 2408.0±459.3 | **2788.2 ± 690.7** | 1469.9 ± 214.1 | 2696.3 ± 810.0 |
> Ant-v4  | 1552.1±237.2 | 1996.6 ± 488.0 | 1453.5 ± 474.4 | **2508.5 ± 492.5** |
> Humanoid-v4 | 99.0±30.2 | 899.4 ± 204.6 | 2469.9 ± 2160.8 | **5161.9 ± 490.9** |
> Reacher-v4 | -8.5 ± 2.2 | -4.6 ± 0.4 | -5.5 ± 1.4 | **-4.1 ± 0.5** |
>
> Comparative evaluation in double gaussian delay distribution
> Environment     | BPQL | State Augmentation-MLP | VDPO | DADAC(Ours)
> -------- | -------- | --------- | -------- | -------- |
> Walker2d-v4  | 517.3 ± 88.3 | 2624.9 ± 911.5 | 1098.1 ± 810.0 | **4693.4 ± 792.1** |
> Hopper-v4  | 752.5 ± 465.9 | 2731.1 ± 446.0 | 611.2 ± 286.0 | **3091.1 ± 284.0** |
> Ant-v4  | 791.2 ± 325.5 | **1788.8 ± 348.0** | 1365.1 ± 531.3 | 1778.8 ± 105.3 |
> Humanoid-v4 | 272.2 ± 162.6 | 714.2 ± 123.2 | 1938.0 ± 122.0 | **4023.9 ± 722.6** |
> Reacher-v4 | -11.7 ± 1.4 | -4.8 ± 1.1 | -7.2 ± 1.0 | **-4.8 ± 0.3** |
>
> Comparative evaluation in uniform delay distribution
> Environment     | BPQL | State Augmentation-MLP | VDPO | DADAC(Ours)
> -------- | -------- | --------- | -------- | -------- |
> Walker2d-v4  | 403.4 ± 116.0 | 1426.8 ± 532.0 | 901.1 ± 222.0 | **4764.6 ± 388.1** |
> Hopper-v4  | 372.7 ± 20.6 | 2442.9 ± 318.0 | 433.3 ± 27.5 | **3116.0 ± 112.7** |
> Ant-v4  | 498.5 ± 314.3 | 1829.7 ± 99.7 | 1282.0 ± 211.1 | **2077.4 ± 44.3** |
> Humanoid-v4 | 146.2 ± 51.4 | 734.6 ± 71.0 | 3622.2 ± 1418.4 | **4195.5 ± 464.9** |
> Reacher-v4 | -13.8 ± 1.8 | **-4.5 ± 0.9** | -10.3 ± 2.4 | -5.4 ± 0.4 |
>
> - [Q5] How does DADAC compare to a delay-free algorithm like robust RL?
>
> Consistent with most of the  related work, we did not consider comparisons with robust reinforcement learning. In my view, if the application scenario (e.g., drone navigation) involves significant delays, selecting DADAC offers a clear advantage, as it more precisely addresses state and control errors caused by delays. On the other hand, if the primary focus is on environmental uncertainties (e.g., parameter perturbations or noise variations) and delay effects are minimal, Robust RL may be preferable for its simplicity and broad adaptability. Nevertheless, I believe that some optimization strategies from Robust RL could be leveraged to enhance the adaptability of delay-aware reinforcement learning.
>
> - [W1] Lack of relevant experiments where both types of delays occur simultaneously
>
> Most related works focus on experiments conducted in environments with observation delays only, and adapting them directly to environments with action delays is often challenging. Therefore, for a fair comparison, we limited our experiments to scenarios with observation delays. To further validate that our method can handle scenarios where both types of delays coexist, we plan to include an additional set of experiments. These will demonstrate that our method can equivalently address environments with a single type of delay as well as those with both types simultaneously.
>
> For fair comparison, delays in single-delay scenarios followed a uniform distribution from 1 to 10, while in dual-delay scenarios (observation and action delays), both followed a uniform distribution from 1 to 5—equivalent to a single delay type with a range of 1 to 10. Due to time constraints, we conducted a limited number of tests, which may lead to performance instability. Nonetheless, current results demonstrate that the proposed DADAC effectively handles environments with both observation and action delays.
>
> Comparison experiments with different types of delays.
> Environment     | DADAC(observation delay) | DADAC(action delay) | DADAC(both observation and action delay) |
> -------- | -------- | --------- | ---------- |
> Walker2d-v4  | 4160.9 ± 171.3 | 4294.9 ± 429.8 | 4072.3 ± 159.0 |
> Hopper-v4  | 2932.3 ± 213.7 | 2850.6 ± 134.9 | 2335.8 ± 187.7 |
> Ant-v4  | 1668.6 ± 189.1 | 1677.0 ± 158.9 | 1443.3 ± 164.4 |
> Humanoid-v4 | 4639.3 ± 917.8 | 3664.1 ± 138.8 | 3864.7 ± 441.3 |
> Reacher-v4 | -5.1 ± 0.7 | -8.6 ± 1.0 | -9.8 ± 0.5 |

---

> > ### Author Response · Authors · 2024-12-03
> >
> > References
> >
> > [1]Bouteiller, Y., Ramstedt, S., Beltrame, G., Pal, C., and Binas, J. Reinforcement learning with random delays. In International conference on learning representations, 2021.
> >
> > [2]Liotet, P., Maran, D., Bisi, L., and Restelli, M. Delayed reinforcement learning by imitation. In International Conference on Machine Learning, pp. 13528–13556. PMLR, 2022.
> >
> > [3]Qingyuan Wu, Simon Sinong Zhan, Yixuan Wang, Yuhui Wang, Chung-Wei Lin, Chen Lv, Qi Zhu, Jürgen Schmidhuber, and Chao Huang. Boosting reinforcement learning with strongly delayed feedback through auxiliary short delays. In Forty-first International Conference on Machine Learning.
> >
> > [4]Qingyuan Wu, Simon Sinong Zhan, Yixuan Wang, Yuhui Wang, Chung-Wei Lin, Chen Lv, Qi Zhu, and Chao Huang. Variational delayed policy optimization. In Advances in Neural Information Processing Systems 37 (2024).
> >
> > [5]Jangwon Kim, Hangyeol Kim, Jiwook Kang, Jongchan Baek, and Soohee Han. Belief projection based reinforcement learning for environments with delayed feedback. Advances in Neural Information Processing Systems, 36:678–696, 2023.
> >
> > [6]Wei Wang, Dongqi Han, Xufang Luo, and Dongsheng Li. Addressing signal delay in deep reinforcement learning. In The Twelfth International Conference on Learning Representations, 2024.
> >
> > [7]Baiming Chen, Mengdi Xu, Liang Li, and Ding Zhao. Delay-aware model-based reinforcement learning for continuous control. Neurocomputing, 450:119–128, 2021.
> >
> > [8]Somjit Nath, Mayank Baranwal, and Harshad Khadilkar. Revisiting state augmentation methods for reinforcement learning with stochastic delays. In Proceedings of the 30th ACM International Conference on Information & Knowledge Management, pp. 1346–1355, 2021.
> >
> > [9]Krasniqi, F., Elias, J., Leguay, J., & Redondi, A. E. (2020, July). End-to-end delay prediction based on traffic matrix sampling. In IEEE INFOCOM 2020-IEEE Conference on Computer Communications Workshops (INFOCOM WKSHPS) (pp. 774-779). IEEE.
> >
> > [10]Wang, Y., Vuran, M. C., & Goddard, S. (2011). Cross-layer analysis of the end-to-end delay distribution in wireless sensor networks. IEEE/ACM Transactions on networking, 20(1), 305-318.
> >
> > [11]Guillo-Sansano, E., Syed, M. H., Roscoe, A. J., Burt, G. M., & Coffele, F.. Characterization of time delay in power hardware in the loop setups. IEEE Transactions on Industrial Electronics, 2020, 68(3), 2703-2713.

---

### Official Review · Reviewer_Ptmx · 2024-10-27

**Soundness:** 3
**Presentation:** 4
**Contribution:** 3
**Rating:** 5
**Confidence:** 4

**Summary:**

This paper proposes a reinforcement learning algorithm for environments with random delays. In real-world scenarios, feedback delays and actuator delays frequently occur due to limited computing resources and bandwidth constraints. Thus, it is crucial to develop a robust algorithm capable of performing well under challenging, low-bandwidth conditions. The proposed algorithm, DADAC, addresses these random delays through two primary components: (1) a Distributional Critic and (2) Delay-Aware Value Correction. The Distributional Critic enhances the agent's robustness by more accurately modeling the uncertainty associated with random delay environments. Meanwhile, Delay-Aware Value Correction adjusts the Bellman equation to account for the probabilistic nature of delays, helping the agent to accurately compute the returns.

**Strengths:**

The paper is well-structured and the contributions are clear.

The novel use of Delay-Aware Value Correction, which applies Bellman updates based on delay-adjusted expectations.

The proposed algorithm shows clear improvement compared to other approaches.

While many conventional methods only handle fixed delays, the proposed algorithm effectively handles random delays, making it a more practical solution.

**Weaknesses:**

Theorem 1 lacks rigorous proof.

More thorough ablation experiments are needed to illustrate how the Delay-Aware Value Correction approach contributes to handling random delays.

Based on my understanding, the proposed method requires prior knowledge of the delay distribution, which may limit its applicability.

**Questions:**

**[Q1]**
In Equation (3), $\gamma$ is expressed with exponential terms for each i. Why is $\gamma$  not expressed this way in Equation (5)?

**[Q2]**
In the proof of Theorem 1, for which distance is the distributional Bellman operator a contraction? Is it KL-divergence, Wasserstein distance, or Cramér distance? Given that your method relies heavily on KL-divergence for the critic loss (Line 200), it is natural to show that the distributional Bellman operator is a contraction with respect to KL-divergence; however, it does not (Bellemare, 2017). Could you elaborate further on the proof of Theorem 1?

**[Q3]**
Why does the "SAC+Value Correction" method perform so poorly on some tasks? It looks like the only difference between DADAC and SAC+Value Correction is whether or not a distributional critic is used. Especially in Humanoid-v4 and Ant-v4, there was no performance improvement at all.

It’s surprising that DSAC and "SAC+Value Correction" show pretty much the same performance across almost all tasks, even though DSAC naively uses the delayed observation. Could you explain why this is happening? It seems like Delay-aware Value Correction does not contribute significantly to DADAC.

**[Minor]**
Typically, many reinforcement learning algorithms conduct experiments using 10 random seeds. Why did you use 8 random seeds instead of 10? Was it to exclude outlier seeds for a more reliable comparison?


[1] Bellemare, Marc G., Will Dabney, and Rémi Munos. "A distributional perspective on reinforcement learning." International conference on machine learning. PMLR, 2017.

---

> ### Author Response · Authors · 2024-11-28
>
> We deeply appreciate your thoughtful review and constructive comments. We have addressed your concerns as follows.
>
> - [Q1] Concerns about inconsistencies in the form of γ
>
> The notation for $\gamma$ in equation (3) is correct; however, due to a typographical error, we omitted the exponential form of $\gamma$ in equation (5). We will address and correct this issue in future versions of the paper.
>
> - [Q2 & W1] Theorem 1
>
> In [1], it has been proven that the distributional Bellman operator is a contraction under the Wasserstein distance. However, the computational complexity of the Wasserstein distance makes it impractical for real-world applications. In distributional reinforcement learning, it is a common setting to approximate the true return distribution using a specific distribution and to define the loss function accordingly [1,2,3,4]. Following the methodology outlined in [2], we adopt Gaussian distribution to approximate the return distribution and employ the KL divergence during the update process. In future versions of this paper, we will enhance the explanation of Theorem 1 and further investigate the convergence properties of our method when employing the KL divergence variant as the distance metric.
>
> - [Q3 & W2] Concerns about ablation studies
>
> Firstly, in our method, distributional returns and the value correction mechanism are complementary and interdependent. Random delays may introduce issues such as missing, duplicate, or out-of-order observations and actions, resulting in sequential feedback from the agent-environment interaction that is inconsistent with the feedback stored in the replay buffer. Due to this inconsistency, SAC+Value Correction suffers from inaccurate Q-value estimation based on sampled data from the replay buffer, thereby reducing the effectiveness of the value correction mechanism.
>
> In contrast, distributional returns can effectively model the uncertainty in delayed environments, enhancing the robustness and multimodal expressiveness of the value function estimation. Consequently, the DSAC method inherently demonstrates a certain level of adaptability to delayed environments, achieving performance comparable to SAC+Value Correction. Furthermore, leveraging the advantage of distributional returns in significantly reducing value function estimation errors, the value correction mechanism becomes more effective. This synergy enables our proposed DADAC method to achieve superior performance in random delay environments.
>
> - [Q4] The number of random seed
>
> In recent related works, we noticed that the number of experimental test runs varies, such as 5 runs (BPQL[5]), 6 runs (DC/AC[6]), 8 runs (State Augmentation-MLP[7]), and 10 runs (AD-RL[8] and VDPO[9]). Due to time constraints, we opted for a moderate number of 8 random seeds and did not exclude any outlier seeds. We have released the complete source code and will include additional experiments in the next version of the paper.
>
> - [W3] Concerns about delay distribution
>
> Our method assumes that the delay distribution is known as prior knowledge and that it possesses stationary properties. In recent related research, most methods typically rely on exact delay information as prior knowledge, such as real-time true delay values [5,6,8,9,10,11,12], and most of them are designed for the enviornment with fixed delays. However, this type of prior knowledge is often difficult to obtain in practical applications and the delays are usually non-fixed in practical systems. This study aims to bring delay-aware reinforcement learning in random delay environments closer to real-world applications.
>
> Some existing work also assumes prior knowledge of the maximum value of random delays [7]，but we found that many real-world delays often follow specific statistical patterns [6,13,14,15], such as network delays. By  leveraging these statistical characteristics as prior knowledge for the delay distribution, our method becomes more suitable for real-world scenarios and outperforms the method with prior knowledge of maximum delay. The analysis of Wi-Fi network delays in [6] shows that the statistical results closely resemble a gamma distribution; in [13], the authors used Gaussian-related distributions to simulate delays. These studies inspired us to design gamma and double Gaussian distributions to simulate various delay environments.
>
> Theoretically, as long as the delay distribution is known, our method can effectively address delay problems, regardless of the specific type of delay distribution. To further validate the adaptability of our method across different delay environments, we have included experimental results based on a uniform distribution with values ranging from 1 to 13.

---

> ### Author Response · Authors · 2024-11-28
>
> **The following are the latest comparison evaluation results.**
>
> Comparative evaluation in gamma delay distribution
> Environment     | BPQL[5] | State Augmentation-MLP[7] | VDPO[9] | DADAC(Ours)
> -------- | -------- | --------- | -------- | -------- |
> Walker2d-v4  | 2450.8 ± 1088.5 | 3761.0 ± 278.6 | 3300.9 ± 976.1 | **4928.5 ± 461.9** |
> Hopper-v4  | 2408.0±459.3 | **2788.2 ± 690.7** | 1469.9 ± 214.1 | 2696.3 ± 810.0 |
> Ant-v4  | 1552.1±237.2 | 1996.6 ± 488.0 | 1453.5 ± 474.4 | **2508.5 ± 492.5** |
> Humanoid-v4 | 99.0±30.2 | 899.4 ± 204.6 | 2469.9 ± 2160.8 | **5161.9 ± 490.9** |
> Reacher-v4 | -8.5 ± 2.2 | -4.6 ± 0.4 | -5.5 ± 1.4 | **-4.1 ± 0.5** |
>
>
>
> Comparative evaluation in double gaussian delay distribution
> Environment     | BPQL[5] | State Augmentation-MLP[7] | VDPO[9] | DADAC(Ours)
> -------- | -------- | --------- | -------- | -------- |
> Walker2d-v4  | 517.3 ± 88.3 | 2624.9 ± 911.5 | 1098.1 ± 810.0 | **4693.4 ± 792.1** |
> Hopper-v4  | 752.5 ± 465.9 | 2731.1 ± 446.0 | 611.2 ± 286.0 | **3091.1 ± 284.0** |
> Ant-v4  | 791.2 ± 325.5 | **1788.8 ± 348.0** | 1365.1 ± 531.3 | 1778.8 ± 105.3 |
> Humanoid-v4 | 272.2 ± 162.6 | 714.2 ± 123.2 | 1938.0 ± 122.0 | **4023.9 ± 722.6** |
> Reacher-v4 | -11.7 ± 1.4 | -4.8 ± 1.1 | -7.2 ± 1.0 | **-4.8 ± 0.3** |
>
>
>
> Comparative evaluation in uniform delay distribution
> Environment     | BPQL[5] | State Augmentation-MLP[7] | VDPO[9] | DADAC(Ours)
> -------- | -------- | --------- | -------- | -------- |
> Walker2d-v4  | 403.4 ± 116.0 | 1426.8 ± 532.0 | 901.1 ± 222.0 | **4764.6 ± 388.1** |
> Hopper-v4  | 372.7 ± 20.6 | 2442.9 ± 318.0 | 433.3 ± 27.5 | **3116.0 ± 112.7** |
> Ant-v4  | 498.5 ± 314.3 | 1829.7 ± 99.7 | 1282.0 ± 211.1 | **2077.4 ± 44.3** |
> Humanoid-v4 | 146.2 ± 51.4 | 734.6 ± 71.0 | 3622.2 ± 1418.4 | **4195.5 ± 464.9** |
> Reacher-v4 | -13.8 ± 1.8 | **-4.5 ± 0.9** | -10.3 ± 2.4 | -5.4 ± 0.4 |
>
> References
>
> [1] Bellemare, Marc G., Will Dabney, and Rémi Munos. A distributional perspective on reinforcement learning. International conference on machine learning. PMLR, 2017.
>
> [2] Jingliang Duan, Yang Guan, Shengbo Eben Li, Yangang Ren, Qi Sun, and Bo Cheng. Distributional soft actor-critic: Off-policy reinforcement learning for addressing value estimation errors. IEEE transactions on neural networks and learning systems, 33(11):6584–6598, 2021.
>
> [3] Nam, D. W., Kim, Y., & Park, C. Y. . GMAC: A distributional perspective on actor-critic framework. In International Conference on Machine Learning (pp. 7927-7936). PMLR.
>
> [4] G. Barth-Maron et al. Distributed distributional deterministic policy gradients. In International conference on learning representations, 2018.
>
> [5] Jangwon Kim, Hangyeol Kim, Jiwook Kang, Jongchan Baek, and Soohee Han. Belief projection based reinforcement learning for environments with delayed feedback. Advances in Neural Information Processing Systems, 36:678–696, 2023.
>
> [6] Bouteiller, Y., Ramstedt, S., Beltrame, G., Pal, C., and Binas, J. Reinforcement learning with random delays. In International conference on learning representations, 2021.
>
> [7] Wei Wang, Dongqi Han, Xufang Luo, and Dongsheng Li. Addressing signal delay in deep reinforcement learning. In The Twelfth International Conference on Learning Representations, 2024.
>
> [8] Qingyuan Wu, Simon Sinong Zhan, Yixuan Wang, Yuhui Wang, Chung-Wei Lin, Chen Lv, Qi Zhu, Jürgen Schmidhuber, and Chao Huang. Boosting reinforcement learning with strongly delayed feedback through auxiliary short delays. In Forty-first International Conference on Machine Learning.
>
> [9] Qingyuan Wu, Simon Sinong Zhan, Yixuan Wang, Yuhui Wang, Chung-Wei Lin, Chen Lv, Qi Zhu, and Chao Huang. Variational delayed policy optimization. In Advances in Neural Information Processing Systems 37 (2024).
>
> [10] Liotet, P., Maran, D., Bisi, L., and Restelli, M. Delayed reinforcement learning by imitation. In International Conference on Machine Learning, pp. 13528–13556. PMLR, 2022.
>
> [11] Baiming Chen, Mengdi Xu, Liang Li, and Ding Zhao. Delay-aware model-based reinforcement learning for continuous control. Neurocomputing, 450:119–128, 2021.
>
> [12] Somjit Nath, Mayank Baranwal, and Harshad Khadilkar. Revisiting state augmentation methods for reinforcement learning with stochastic delays. In Proceedings of the 30th ACM International Conference on Information & Knowledge Management, pp. 1346–1355, 2021.
>
> [13] Krasniqi, F., Elias, J., Leguay, J., & Redondi, A. E.. End-to-end delay prediction based on traffic matrix sampling. In IEEE INFOCOM 2020-IEEE Conference on Computer Communications Workshops (INFOCOM WKSHPS) (pp. 774-779). IEEE.
>
> [14]Wang, Y., Vuran, M. C., & Goddard, S. (2011). Cross-layer analysis of the end-to-end delay distribution in wireless sensor networks. IEEE/ACM Transactions on networking, 20(1), 305-318.
>
> [15]Guillo-Sansano, E., Syed, M. H., Roscoe, A. J., Burt, G. M., & Coffele, F.. Characterization of time delay in power hardware in the loop setups. IEEE Transactions on Industrial Electronics, 2020, 68(3), 2703-2713.

---

### Meta-Review · Area_Chair_u4v6 · 2024-12-16

**Metareview:**

This works proposes deep reinforcement learning algorithms to deal with random delays in reinforcement learning. In order to achieve this they model delay uncertainty by proposing a distributional deep reinforcement learning algorithm that can represent the distribution of this Q values. Concerns remained after the discussion period regarding missing baselines as well as a lack of discussion / explanation of some of the methods proposed.

**Additional Comments On Reviewer Discussion:**

This works proposes deep reinforcement learning algorithms to deal with random delays in reinforcement learning. In order to achieve this they model delay uncertainty by proposing a distributional deep reinforcement learning algorithm that can represent the distribution of this Q values. The reviewers agreed the approach proposed in this work holds promise, but concerns about the limited nature of the experiments in the form of missing benchmarks such was raised. Moreover, there were comments about the lack of discussion about the correction mechanism on the distribution of return, pointing to the lack of discussion about it.

---

### Decision · Program_Chairs · 2025-01-22

Reject